# Component-Resolved Evaluation of the Risk and Success of Immunotherapy in Bee Venom Allergic Patients

**DOI:** 10.3390/jcm11061677

**Published:** 2022-03-17

**Authors:** Marta Rosiek-Biegus, Robert Pawłowicz, Agnieszka Kopeć, Magdalena Kosińska, Marta Wrześniak, Marita Nittner-Marszalska

**Affiliations:** Department of Internal Medicine, Pneumology and Allergology, Wroclaw Medical University, 50-369 Wroclaw, Poland; robert.pawlowicz@umed.wroc.pl (R.P.); agnieszka.kopec@umed.wroc.pl (A.K.); magdalena.kosinska@umed.wroc.pl (M.K.); marta.wrzesniak@umw.edu.pl (M.W.); marita.nittner-marszalska@umed.wroc.pl (M.N.-M.)

**Keywords:** bee venom allergy, venom immunotherapy, component-resolved diagnostics

## Abstract

Venom immunotherapy (VIT) is the only efficient therapy for the Hymenoptera insect venom allergy. Immunotherapy with bee venom is encumbered with a higher risk of systemic side effects and/or therapeutic failures. The objective of the study was to assess if specific profiles of molecular IgE (Immunoglobulin E) responses are associated with an increased risk of systemic side effects and/or the treatment’s inefficacy. The study group numbered 64 bee venom allergic patients (BVA) who received venom immunotherapy modo ultra-rush (VIT-UR), (f/m: 32/32, mean age 43.4 ± 17.2). In total, 54.84% of them manifested allergic reactions of grades I-III (acc. to Mueller’s scale), while 48.66% manifested reactions of grade IV. In all the patients, IgE against bee venom extract, rApi m 1 and tryptase (sBT) were assessed. In 46 patients, assessments of IgE against rApi m 2, 3, 5, 10 were also performed. BVA patients manifesting cardiovascular symptoms (SYS IV^0^) showed higher levels of both sIgE-rApi m 5 (*p* = 0.03) and tryptase (*p* = 0.07) than patients with SYS I–III. Systemic adverse events during VIT with bee venom were more frequent in the induction phase than in the maintenance phase: 15.22% vs. 8.7%. In BVA patients who experienced systemic adverse events during VIT, higher concentrations of sIgE-rApi m 5 (*p* < 0.05), rApi m 1 (*p* = 0.009), and sBT (*p* = 0.019) were demonstrated. We conclude that higher levels of sIgE against rApi m 1, rApi m 5, and tryptase many constitute a potential marker of the severity of allergic reactions and therapeutic complications that can occur during VIT with bee venom.

## 1. Introduction

Hymenoptera venom allergy (HVA), mainly wasp (YJVA) and bee venom (BVA), is manifested by mild local reactions (HVA-LL) and much more serious systemic reactions that are life-threatening (HVA-SYS). HVA is now recognized as one of the most common causes of anaphylaxis in both adults and children [1,2]. Local manifestations of HVA, called large local reactions (HVA-LL), are reported in over a dozen percent of the population (2.4–26.4%) [2]. The frequency of systemic reactions (HVA-SYS) is 10 times less common than the local ones [3] (up to 7.5% in adults and 6.5% in children) [4,5]. BVA causes special diagnostic and therapeutic problems, which is due to the lower sensitivity of the diagnostic procedures and the fact that venom immunotherapy (VIT) is both less safe and less effective.

HVA diagnostics is based, in the first stage, on the medical history of the allergic reaction, skin tests with venom extracts, and the determination of specific IgE against extracts of both venoms. The sensitivity of diagnostic tests with the extract is lower in BVA than in YJVA and amounts to 91% for intradermal tests (IDT) with HBV, and 96% for YJV. The sensitivity of serological tests is 90–100% (BVA) and 83–97% (YJVA) [6,7]. The disadvantage of the extract as a diagnostic material is the potential quantitative and qualitative deficiencies of individual allergen molecules, as well as the addition of non-allergenic substances with irritating properties.

The development of biological techniques allows purified allergen components to be obtained from venom extract, as well as the production of the venom’s native components and recombinant molecules [8]. Diagnostics based on the assessment of the presence of sIgE in serum against individual allergen molecules (components) of the venom extract (component-resolved diagnosis (CRD)) plays an increasingly important role in the diagnosis and treatment of HVA. This technique increases diagnostic precision and is a recognized step in HVA diagnostics in the case of negative venom extract test results or double (bee and wasp) allergies [9]. Diagnostic sensitivity with the use of available wasp venom molecules (Ves v 1 and Ves v 5) is higher than in the case of the bee venom allergy (with the use of 5 molecules: Api m 1, 2, 3, 5, 10) and is equal to 95% and 75%, respectively [9].

The only treatment of HVA is VIT. It is a highly effective method, especially in patients undergoing immunotherapy with wasp venom and provides protection in 91–96% of them. However, the efficacy and safety of VIT in BVA is lower. Therapeutic failures are observed in 16–25% of patients treated with BVA, and allergic side effects of this treatment occur 3–6 times more often than in patients undergoing VIT with wasp venom [10,11,12].

The reason for the discrepancy in the results of the diagnostic sensitivity of BVA and YJVA, as well as in the results of treatment with these venoms is still unknown and may be related to the complexity of the bee venom. To date, 12 bee venom allergens have been described (Table 1). The complexity of the allergen composition of BVA is reflected in the allergy profiles to this venom.

Only a few studies concerning BV allergy profiles have been carried out. Moreover, no research has been conducted regarding the correlation between the allergy to BV molecules, the severity of the post-stinging reaction, and the risk/effectiveness of HBV immunotherapy. The role of the connection between allergy to Api m 10 and the risk of VIT failure is increased—a similar relationship has not been demonstrated for other molecules. However, it cannot be ruled out that the link between an allergy to Api m 10 and VIT failure may be the result of a reduced representation of this molecule in the vaccines of some manufacturers [21]. The problem of instability and increased degradation of Api m 10 in the venom extract is also important [22].

In the presented study, we tried to assess whether specific profiles of molecular IgE responses are associated with an increased risk of systemic side effects or treatment inefficacy.

## 2. Material and Methods

### 2.1. Study Protocol

Sixty-four adult BVA patients aged 43.4 ± 17.2 years with a history of a systemic allergic reaction after bee stings of grade I-IV, who had positive intradermal tests (IDT) to bee venom extract of less than or equal to 10–4 g/L, and BV extract-specific serum IgE levels of 0.35 kU/L or greater, were included in the study (Table 2). The IDTs were performed with a volume of 0.02 mL of an aqueous solution of BV extract (Pharmalgen ALK-Abello, Hørsholm, Denmark). In all 64 patients, sIgE against rApi m 1 molecules and serum tryptase (sBT) level were determined. Additionally in 46 individuals in this group (46/64) rApi m 2, rApi m 3, rApi m 5 and rApi m 10 sIgE were determined. Both groups of patients were clinically and immunologically representative when compared to the entire group of patients qualified for BV-VIT in our center in the years 2010–2019 (*n* = 232).

Patients were qualified to VIT modo ultra-rush (VIT-UR) according to the European Academy of Allergy and Clinical Immunology (EAACI) guidelines [10]. The exclusion criteria in this study were: pregnancy, breastfeeding, severe psychiatric disease, unstable respiratory and cardiovascular diseases, immunosuppression, immunodeficiency, and if an active autoimmune process and cancer had been diagnosed in the last five years. During the VIT-UR, the patients received 6 injections of BV starting at 0.1 mcg (day 1), with a top dose of 50 mcg and a cumulative dose of 111.1 mcg. During the initial phase, the patients’ blood pressure and pulse were monitored, and they were subjected to electrocardiography. Moreover, venous access for an infusion of sodium chloride 0.9% and the infusion’s peak flow were established before the first injection. On day 8, they received 2 injections of 50 mcg, and on day 22, they received 1 injection of 100 mcg. After the maintenance phase, further injections of the maintenance dose of 100 mcg of BV were administered. In the build-up phase of VIT, an aqueous solution of Pharmalgen ALK-Abello venom was used, and in the maintenance phase of VIT, the Alutard SQ ALK-Abello venom depot vaccine was used.

Written informed consent was obtained from each patient. The study protocol was approved by the Ethical Committee of the Medical University of Wrocław, Poland. The study was conducted as part of the first author’s PhD thesis.

### 2.2. Assessment of Systemic Allergy Side Effects

Systemic allergic side effects during the initial phase of VIT and also during the first 6 months of the maintenance phase were classified according to the Mueller scale in grades I to IV (Table 3) [23]. VIT was considered a failure if there was the occurrence of an allergic reaction in the course of each of its phases.

### 2.3. Immunologic Analyses

To evaluate the sIgE with the venom extract, determination of the sIgE BV extract was performed using the ELISA method. The cut-off point for the positive result was 0.35 kUA/L (Table 4). Determination of the sIgE BV allergen components was performed using the ImmunoCAP Phadia100 system measurements (Thermo Fisher Scientific Inc., Göteborg, Sweden) by the ELISA method. In addition to the standard curve (ImmunoCAP Specific IgE Calibrators No: 10-9460-01), controls were performed during each analysis (ImmunoCAP Specific IgE No: 10-9462-01). The IgE concentration against the following recombinant components in bee venom was determined: rApi m 1 *Phospholipase A2*, Honey bee: (No: 10 14-4987-01); rApi m 2 *Hyaluronidase*, Honey bee: (No: 14-6014-01); rApi m 3 *Acid phosphatase*, Honey bee: (No: 14-6015-01); rApi m 5 *Dipeptidyl peptidase*, Honey bee: (No: 14-6016-01); and rApi m 10 *Icarapin*, Honey bee: (No:14-6004-01).

To evaluate tryptase, sBT was determined using the ImmunoCAP Phadia100 system measurements (Thermo Fisher Scientific Inc.). The sBT result <11.4 ng/mL was considered to be correct.

### 2.4. Statistical Analysis

Continuous variables with a normal distribution were described using means ± standard deviation (SD), variables with skewed distribution were described by medians with upper and lower quartiles, and categorized variables were given as numbers and percentages. The statistical significance of differences between the groups was assessed using the *t*-test, Mann–Whitney U-test, and Chi2 test, where appropriate. The *p* < 0.05 was considered statistically significant. Statistical analyses were performed using STATISTICA 12 (StatSoft).

## 3. Results

### 3.1. Allergy Profile

In the studied group, allergy to the molecule rApi m 1 was the most common and was demonstrated in 62.5% of the individuals. Sensitization to at least one molecule of the examined panel was found in 82.61% (38/46) of patients; allergy to rApi m 10 was 58.7%, to rApi m it was 2 36.96%, to rApi m it was 5 32.61%, and to rApi m 3 it was 30.43%. Among the BVA patients (by the cut-off point of 0.35 kUA/L), 21 allergy profiles were detected (Figure 1).

In 17.39% (8/46) of the examined patients, sensitization to any of the examined molecules was not confirmed. Shifting the positive cut-off point to 0.1 kUA/L confirmed sensitization in an additional three patients: three to rApi m 1, and in one of these patients also to rApi m 10. When the cut-off point was shifted to 0.1 kU/L, the percentage of patients who were allergic to the extract, but not allergic to the examined molecules of bee venom, was 10.87%.

### 3.2. Profile of Molecular Sensitization and the Severity of the HVA Reaction

Patients with a systemic reaction with cardiovascular involvement grade IV (Table 5) had higher median concentrations of sIgE antibodies against the rApi m 5 molecule (median [Q25–Q75]: 0.50 [0.01–1.24] vs. 0.04 [0.0–0.12], *p* = 0.03) than patients with non-cardiovascular reactions. There was no correlation between the severity of the SYS IV reaction and the concentrations of antibodies to the remaining tested molecules: rApi m 1, rApi m 2, rApi m 3, or rApi m 10. Patients with SYS IV reactions also had higher levels of sBT mean ± SD (8.7 ± 5.0 vs. 5.2 ± 3.0, *p* = 0.007) compared with patients experiencing milder severity of the reaction (Table 5).

### 3.3. VIT Allergic Side Effects and the Molecular Profiles of Allergies

During VIT-UR, seven patients demonstrated a systemic allergic reaction (SYS I-57.14%, SYS II-28.57%, SYS III-0%, SYS IV-14.28%). Patients with allergic side effects during VIT-UR had higher levels of antibodies against rApi m 1 (*p* = 0.009) and rApi m 5 (*p* = 0.02) (Table 6). A trend towards higher concentrations of IgE-rApi m 10 (*p* = 0.06) was also demonstrated in the group with allergic side effects than in patients without allergic side effects in this phase of the treatment. There were no differences in the concentrations of rApi m 2 and rApi m 3 in the group of patients with allergic side effects during VIT-UR and the group without allergic side effects in this phase of VIT.

Patients with a bee venom allergy and allergic side effects during VIT-UR had a higher sBT concentration of 8.51 ± 5.62 vs. 5.55 ± 3.42 *p* = 0.019 than patients with uncomplicated VIT-UR.

Patients with allergic side effects during the entire course of VIT showed a trend towards higher rApi m 5 values when compared to the group without allergic side effects: median [Q25–Q75] 0.41 [0.01–1.12] vs. 0.04 [0–0.50] *p* = 0.07 (Table 7). Moreover, both groups differed in the classes’ distribution of sIgE-rApi m 5. In patients without allergic side effects during the entire period of immunotherapy, there was a higher percentage of patients with sIgE-rApi m 5 concentrations below the cut-off point (*p* = 0.02). There were no differences between the concentration of sIgE against rApi m 1, 2, 3, or 10 between the group with allergic side effects and the group without allergic side effects during VIT.

## 4. Discussion

In the presented study, the analysis of the relationship between sensitization to five tested BV allergen molecules, and also the severity of the systemic sting reaction and the relationship between the sensitization profile and the safety of immunotherapy in patients with HBV allergy, was undertaken.

In our study, we discovered a higher concentration of sIgE-rApi m 5 (*p* = 0.03) in patients with a history of HBV allergy with cardiovascular involvement (SYS IV), which may indicate an association of allergy to this molecule with the most severe HVA systemic reactions. Observations regarding the relationship between the severity of HBV allergy and the molecular profile are random [24].

In the study Arzt et al. [25] conducted on 134 patients allergic to bee venom, no relationship was observed between allergy to HBV molecules (including Api m 5) and the severity of HVA. However, the comparison of the results of both studies is hampered by the lack of data regarding the size of the group of patients with the most severe SYS IV reaction in the Arzt study, as well as the percentage of allergy to Api m 5, which in the Austrian patient population seems to be generally lower than in our group. Ruiz et al. indicate a relationship between the severity of the HVA reaction and the concentrations of sIgE-rApi m 4 and sIgE-rApi m 1 (sIgE-rApi m 5 was not determined) [26]. In this study, patients (*n* = 31) were divided into two groups depending on the concentration of sIgE-rApi m 4 (the cut-off point was 0.98 kU/L). In the group of patients with levels of sIgE-rApi m 4 higher than 0.98 kU/L, more severe systemic reactions following a stinging were observed. In this study, a low number of patients with cardiovascular symptoms (*n* = 5) were considered which means that this result requires confirmation in a larger group of patients. In our study, we did not show any correlation between the severity of the systemic reaction with cardiovascular involvement (SYS IV) and the concentration of antibodies against the remaining tested molecules: rApi m 1, rApi m 2, rApi m 3, and rApi m 10.

For several years, there has been a debate about the relationship between the sensitization profile and the safety and effectiveness of HBV immunotherapy. Both of these phenomena seem to be related: in patients with allergic side effects during VIT, immunotherapy is more often ineffective, which results in the postulate of prolonging the duration of VIT. In the context of our results, higher concentrations of sIgE against the molecules rApi m 1 and rApi m 5 and potentially also rApi m 10 (trend), may be associated with a higher incidence of allergic side effects during VIT-UR. In the discussion on the relationship between the allergy profile and the course of VIT, the results of the study by Frick et al. [21] are also important. They demonstrated the failure of VIT in patients with predominant allergy to rApi m 10. The same study also showed a low representation of Api m 10 in the bee venom vaccine used during immunotherapy. These results correlate with the data of Blanck et al., who demonstrated the deficiency of rApi m 3 and rApi m 10 molecules in bee venom vaccines from various producers [22]. Taking into account that rApi m 1 and rApi m 4 are the only significant components in lyophilized venom, and that the remaining molecules constitute only 0.2–6% (venom dry weight), their deficiency in the production of venom vaccines is likely.

Disturbance of the proportions or even the lack of individual molecules may also explain the lower effectiveness of bee venom immunotherapy than wasp venom immunotherapy. In wasp venom, there are two molecules Ves v 1 and Ves v 5, which constitute 6–14% and 5–10% of the dry venom weight [27]. The dose of wasp venom administered during immunotherapy (100 mcg) is many times higher than the dose of wasp venom administered by the insect during a sting. This implies that both components should be in the right amounts in wasp venom vaccines. In contrast, the dose of bee venom in a bee venom vaccine is approximately the same as the dose administered during a bee sting. A deficiency of one of the components, especially those present in venom in very low concentrations, may adversely affect the effectiveness of bee venom immunotherapy.

The results of the study showed that 82.61% of HBV allergic patients showed sensitization to at least one of the tested molecules, with the most common was sensitization to rApi m 1. The diagnostic usefulness of individual bee venom molecules is, therefore, lower than in wasp venom. This is due to the complexity and diversity of molecular sensitization profiles in the population of patients with HVA to bee venom, which was demonstrated in the study. Our patients presented 21 different molecular sensitization profiles when assessing sIgE against five molecules of bee venom. The number of profiles increased with the number of molecules tested. However, the variation in sensitization profiles may be more significant. Kohler et al., when assessing sIgE against 6 molecules in a study of 144 patients with BV allergy showed as many as 39 different profiles [16]. This multitude of profiles and the lack of an adequate representation of molecules in bee venom vaccines used during immunotherapy may also be one of the factors responsible for the lower effectiveness of bee venom immunotherapy than the much less complex wasp venom.

In our study, we also observed a high percentage (17.39%) of patients with confirmed allergy to the venom extract, and unconfirmed allergy to the tested venom allergen components. There may be two explanations for this. First, the concentration of sIgE against the venom components may be too low in relation to the sensitivity of the current methods of their detection. Second, the lack of sIgE against the test molecules may reflect an allergy to “new” as yet unidentified “smaller” bee venom allergens (e.g., Api m 11 or Api m 12). In these patients, the potential efficacy of VIT may be lower due to the lack of sufficient allergenic molecules in the vaccine. Determining sIgE against a higher number of molecules increases the diagnostic value of the procedure.

A high concentration of sBT is a known, unmodified risk factor for both the severity of the post-stinging reaction and VIT failure [11,28]. In the presented study, the results of tryptase concentration are compatible with previous reports. We observed higher levels of tryptase in patients with SYS IV systemic reactions (*p* = 0.007), in the group of patients with complications during the VIT -UR induction phase (*p* = 0.019), and also in the course of the whole process of immunotherapy (*p* = 0.03).

The study has several limitations that have to be mentioned. First, for the molecular sensitization profiles under investigation in BVA patients, sensitization to five molecules was assessed. However, neither rApi m 4, nor the other known argon molecules of bee venom were included in the study. A possible assessment of the whole aggregate spectrum of the venom is limited by the insufficient commercial availability of the reagents necessary for the laboratory assay of all known components of bee venom. Second, the optimal method to test the efficacy of VIT with bee venom is a sting challenge. The protocol of this study did not involve this method. Instead, the inefficacy of VIT was assumed on the basis of the occurrences of adverse events during the induction and/or maintenance phase of the treatment. Finally, the study was conducted on a small group of patients (*n* = 64). For this reason, further studies involving a bigger group of patients are needed.

## 5. Conclusions

In conclusion, it can be stated that higher levels of sIgE against rApi m 1, rApi m 5, and tryptase may constitute a potential marker of the severity of allergic reactions and therapeutic complications that can occur during VIT with bee venom.

## Figures and Tables

**Figure 1 jcm-11-01677-f001:**
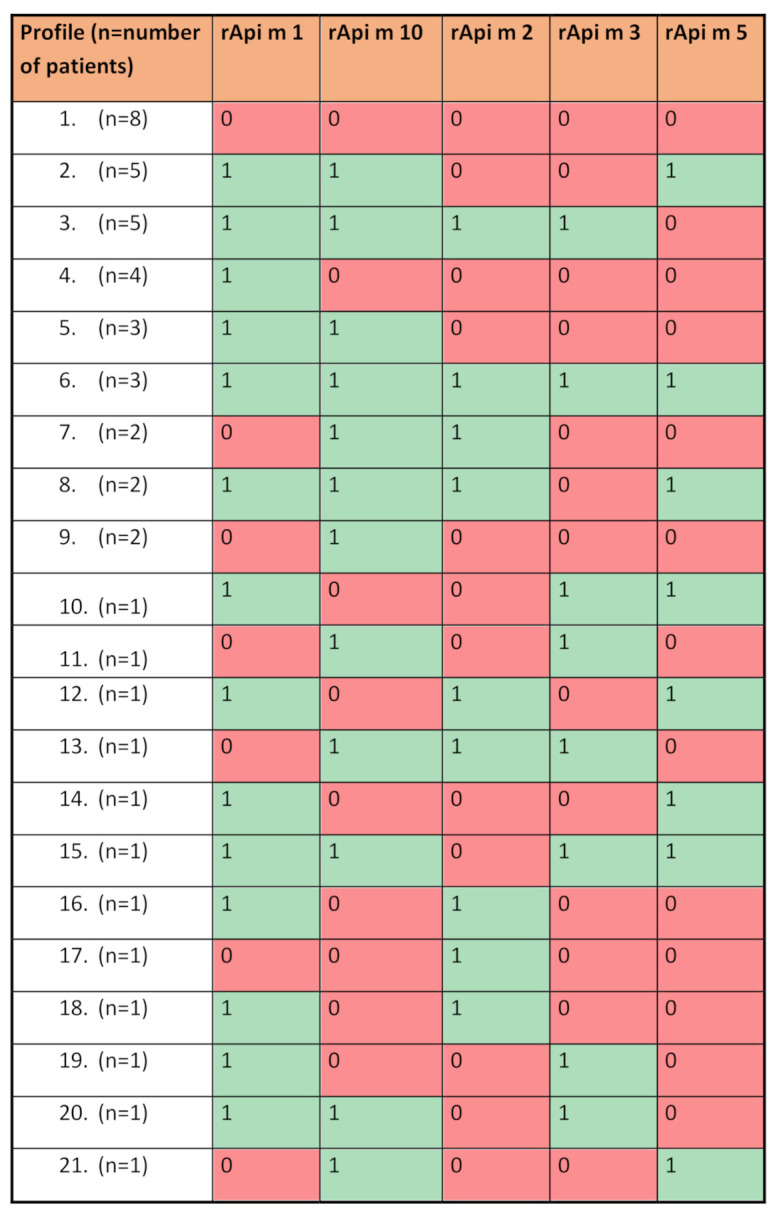
Profile of allergies to molecules of bee venom Api m 1, 2, 3, 5, and 10 in the group of patients allergic to bee venom (N = 46). 1—presence of allergy; 0—no allergy.

**Table 1 jcm-11-01677-t001:** Bee venom allergens.

Biochemical Name	Allergen
Main HBV allergens	
phospholipase A2	Api m 1 [13]
acid phosphatase	Api m 3 [14]
melittin	Api m 4 [15]
icarpin	Api m 10 [14]
Minor HBV allergens	
hyaluronidase	Api m 2 [16]
dipeptidyl peptidase IV	Api m 5 [16]
serine protease inhibitor	Api m 6 [16]
serine protease 1	Api m 7 [17]
carboxyesterase	Api m 8 [17]
carboxypeptidase serine	Api m 9 [18]
MRJP proteins 8 and 9	isoallergens Api m 11.0101 and Api m 11.0201 [19]
vitalogenin	Api m 12 [20]

**Table 2 jcm-11-01677-t002:** Demographic and clinical characteristics (64 patients vs. 46 patients).

HBV Allergy (*n*)	Sex m/f	Age Mean ± SD	Systemic Reaction *n* (%)	sIgE Median [Q25–Q75]
*n* = 64	32/32	43.4 ± 17.2	SYS I-2 (3.13%)	5.3 [1.5–26.5]
			SYS II-4 (6.25%)	
			SYS III-28 (43.75%)	
			SYS IV-30 (46.88%)	
*n* = 46	23/23	43.5 ± 17.2	SYS I-1 (2.17%)	5.64 [1.7–49.5]
			SYS II-4 (8.7%)	
			SYS III-20 (43.48%)	
			SYS IV-21 (45.65%)	
*p*	*p* = 1	*p* = 0.97	*p* = 0.96	*p* = 0.56

**Table 3 jcm-11-01677-t003:** Classification of anaphylactic reaction after insect stings according to Mueller [23].

**SYS I**	Generalized urticaria, itching, malaise, and anxiety.
**SYS II**	Any of the above plus two or more of the following: Generalized oedema; constriction in the chest; wheezing; abdominal pain, nausea and vomiting; and dizziness.
**SYS III**	Any of the above plus two or more of the following: Dyspnoea; dysphagia; hoarseness or thickened speech; confusion; and feeling of impending disaster.
**SYS IV**	Any of the above plus two or more of the following: Cyanosis; fall in blood pressure; collapse incontinence; and unconsciousness.

**Table 4 jcm-11-01677-t004:** Concentration range of sIgE classes.

sIgE Class	[kUA/L]
0	<0.35
1	0.35–0.70
2	0.71–3.50
3	3.51–17.50
4	17.51–50.00
5	50.01–100.0
6	>100

**Table 5 jcm-11-01677-t005:** Comparison of serum levels of specific IgE against bee venom allergen molecules and basal tryptase among patients with reactions to SYS IV and SYS I, II, or III.

Variable	Reaction SYS IV *n* = 21 *	Reaction SYS I–III *n* = 25 *	*p*-Value
sIgE rApi m 1 (kUA/L)	1.66 [0.38–4.13]	0.43 [0.12–3.67]	0.44
sIgE rApi m 2 (kUA/L)	0.13 [0.02–1.8]	0.1 [0.01–0.94]	0.83
sIgE rApi m 3 (kUA/L)	0.2 [0.001–1.11]	0.15 [0.01–0.47]	0.62
sIgE rApi m 5 (kUA/L)	0.5 [0.01–1.24]	0.04 [0.0–0.12]	0.03
sIgE rApi m 10 (kUA/L)	1.01 [0.11–3.71]	0.5 [0.06–1.4]	0.28
tryptase (ng/mL)	8.7 ± 5.0	5.2 ± 3.0	0.007

* The entire molecule profile (Api m 1, Api m 2, Api m 3, Api m 5, and Api m 10) was determined in a group of 46 patients. Data presented as the median [Q25–Q75] or mean ± SD.

**Table 6 jcm-11-01677-t006:** Comparison of serum levels of specific IgE against bee venom allergen molecules among patients with and without allergic side effects in the induction phase of bee venom immunotherapy.

Variable	VIT-UR-with SYS	VIT-UR-without SYS	*p*-Value
sIgE rApi m 1 (kUA/L) *	3.9 [0.93–9.43]	0.43 [0.07–2.86]	0.009
sIgE rApi m 2 (kUA/L) **	0.3 [0.03–2.74]	0.08 [0.01–1.94]	0.24
sIgE rApi m 3 (kUA/L) **	0.6 [0.2–1.52]	0.08 [0.001–0.37]	0.09
sIgE rApi m 5 (kUA/L) **	1.65 ± 2.8	0.39 ± 0.72	0.02
sIgE rApi m 10 (kUA/L) **	1.54 [0.76–4.61]	0.45 [0.04–1.74]	0.06

* *n* = 64, ** *n* = 46, Data presented as the median [Q25–Q75] or mean ± SD.

**Table 7 jcm-11-01677-t007:** Comparison of serum levels of specific IgE against bee venom allergen molecules and basal tryptase among patients with and without allergic side effects during the bee venom immunotherapy.

Variable	VIT without SYS	VIT with SYS	*p*-Value
sIgE rApi m 1 (kUA/L) *	0.64 [0.12–3.67]	2.25 [0.32–6.02]	0.42
sIgE rApi m 2 (kUA/L) **	0.09 [0.01–0.94]	0.14 [0.01–1.63]	0.94
sIgE rApi m 3 (kUA/L) **	0.15 [0.01–0.47]	0.14 [0.01–0.86]	0.89
sIgE rApi m 5 (kUA/L) **	0.04 [0.00–0.50]	0.41 [0.01–1.12]	0.07
sIgE rApi m 10 (kUA/L) **	0.52 [0.06–1.74]	1.11 [0.07–3.29]	0.55
tryptase (ng/mL)	5.35 ± 3.23	7.68 ± 5.12	0.03

* *n* = 64, ** *n* = 46, Data presented as the median [Q25–Q75] or mean ± SD.

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
