# Peer review of "Component-Resolved Evaluation of the Risk and Success of Immunotherapy in Bee Venom Allergic Patients"

_jcm, 2022, doi:10.3390/jcm11061677_

Round 1

Reviewer 1 Report

In their manuscript, Rosiek-Biegus et al. evaluate the IgE profiles towards 5 allergenic bee venom molecules among patients suffering from honeybee venom allergy and undergoing allergen immunotherapy (VIT). The authors included 64 patients in the analysis in order to evaluate, whether specific profiles of molecular IgE responses are associated with an increased risk of systemic side effects or treatment inefficacy. This research question has been previously addressed by other groups, however, regional differences may be of interest and knowledge on the matter is heterogeneous, so the manuscript is of interest for the scientific community. However, quite a few open points need to be addressed very carefully as indicated below. Further, the manuscript MUST be checked by a native English speaker as it contains a high number of grammar and spelling mistakes.

Major comments:

  • The manuscript would benefit from a clear hypothesis. Also in the abstract, the objective and aims of the study need to be made clear. In the discussion, the reader should learn whether the hypothesis has been confirmed or not. Is the hypothesis that severe systemic reactions and VIT side effects are related to specific sensitization profiles? This is unfortunately not presented very clearly.
  • It is not clear why the authors include 64 patients in the analysis if only 46 patients have the necessary complete data set to assess sensitization profiles. Depending on the main hypothesis, the study population needs to be chosen adequately. If there is a good reason to include 64 patients but then only test 46, the authors need to explain well and assess any important differences between the two groups (as done in table 1 for the study population versus general patient population. Also for the results it is often note clear whether they relate to 64 or 46 patients.
  • Essential information on the in- and exclusion criteria of participants are missing. This must be included in the methods section.
  • It should be mentioned how the tested molecules (5) have been chosen. What were the selection criteria? Why was Api m 4 not tested? This should be added to the methods and is far more important than the specific performance details of the intradermal tests which are not essential for the point of the paper.
  • There is no information whether the molecules were recombinant or native. Please add this information as it is essential for the discussion of the results.
  • The tables are far to extensive and their number should be reduced. All tables and figures need extensive revision (see minor comments for concrete recommendations).
  • It is important to correctly distinguish “allergy” from “sensitization” throughout the entire manuscript. For example in line 79, the authors should say “sensitization profile”
  • The manuscript needs careful revision by an English native speaker. Many sentences are unnecessarily complex and there are many grammar and spelling mistakes.
  • A section on limitations is missing.

Minor comments:

Lines 9 ff Abstract: The abstract should start with a short introductory sentence on the general background.

Abbreviations should be fully described when first mentioned. After that, only the abbreviation should be mentioned. It is unnecessary to introduce abbreviations (like in the first sentences of the introduction) if these terms are then not used later in the manuscript. There should be a thorough list of abbreviations as not all of them have been introduced and are simply used, so the reader does not know what they stand for.

The introduction is too detailed. Lines 60-69 could be removed and instead the information compiled in an informative table 1.

Recommendations for all tables: headers are far too long. Use abbreviations and remove non-essential information. Pay attention to correct spelling (e.g. in table 1 it should be severity of allergic reaction) and proper wording (e.g. sex is usually female/male and not w/m). Median should not be abbreviated Me as the word “Mean” also starts with the same letters. It needs to be clear for every number in a table what it described, including indication of units (e.f. what are the values in table 1 column 4?).

Table 1 should be removed. The results can be mentioned in the text very shortly. Instead, the information on allergenic molecules should rather be sumarized in a table than in a long text passage.  

Table 2 should be moved to the e-repository

Table 3 should be removed, this information is not essential for the manuscript.

Table 4 should be moved to the e-repository

Figure 1: as some sensitization profiles are repeated in several patients this figure should be condensed. Instead of indicating 46 profiles, only the 21 distinct profiles should be shown and a column should be added showing how many patients had the respective profile. Ideally the profiles should then be sorted top down according to their prevalence.

Tables 5,6 and 7 should be combined in one table. Headers need to be shortened by removing repetitive information and moving general aspects to the legend. You may also work with * to add specific information.

Table 8 : the table should be deleted. Data here collected should be explained in the results.

References: make sure to use correct reference styles. For example, the journal in Ref 9 is not indicated correctly.

Tryptase results are not discussed. As they fit previous knowledge, no large discussion is needed, but they should at least shortly be mentioned in the context of other researchers´work.

How do the authors explain the rather high rate of patients with no IgE response at all or only reacting to bee venom extract? Please take cross-reactivity into consideration and discuss this aspect.

Author Response

Thank you very much for your comment.

Major comments:

  1. The manuscript would benefit from a clear hypothesis. Also in the abstract, the objective and aims of the study need to be made clear. In the discussion, the reader should learn whether the hypothesis has been confirmed or not. Is the hypothesis that severe systemic reactions and VIT side effects are related to specific sensitization profiles? This is unfortunately not presented very clearly.

Ad 1. Thank you for your remarks. The objective of the study was to assess the usefulness of molecular diagnosis in predicting the safety of VIT performed with bee venom. The main hypothesis was to determine “whether specific profiles of molecular IgE responses are associated with an increased risk of systemic side effects or treatment inefficacy?” The hypothesis has been specified in both the abstract and the introduction to the paper. Fragments of the discussion have been revised so that the study’s results could be clearly commented on.

  • It is not clear why the authors include 64 patients in the analysis if only 46 patients have the necessary complete data set to assess sensitization profiles. Depending on the main hypothesis, the study population needs to be chosen adequately. If there is a good reason to include 64 patients but then only test 46, the authors need to explain well and assess any important differences between the two groups (as done in table 1 for the study population versus general patient population. Also for the results it is often note clear whether they relate to 64 or 46 patients.

Ad 2. Thank you for your comment. SIgE antibodies against Api m 1 were assessed in 64 individuals. SIgE against a panel of molecules Api m 2, 3, 5,10 were assessed in 46 of them. For technical reasons, sIgE assessment against the entire panel in the whole study group was not possible. Even so, we decided to present the results of the assessment of the presence of sIgE against  Api m 1 in 64 individuals in order that the analysis of the involvement of sIgE against this molecule could be of carried out on a larger sample of subjects. Following the reviewer’s suggestion, the group of 18 patients (sIgE against Api m 1 alone) was compared with the group of 46 patients (sIgE against the whole molecular panel) as done in table 1 for the study population vs. the general patient population. No significant differences between the two groups were found.

3, Essential information on the in- and exclusion criteria of participants are missing. This must be included in the methods section.

Ad 3. Thank you for your comment. The patients who were recruited to the study met qualification criteria for venom immunotherapy defined by EAACI, i.e. they presented clinical life threatening symptom of allergy to Hymenoptera  sting, they had positive results of skin tests and positive sIgE against insect venom. Contraindications to VIT constitute exclusion criteria. The information about exclusion criteria has been added to the text.

  1. It should be mentioned how the tested molecules (5) have been chosen. What were the selection criteria? Why was Api m 4 not tested? This should be added to the methods and is far more important than the specific performance details of the intradermal tests which are not essential for the point of the paper.

Ad 4. The selection of molecules for the study was determined by two substantive factors The former relates to the study of Köhler et al who  calculated the contribution of sIgE to single honeybee venom allergens to sIgE reactivity to whole honeybee venom. It was shown that there is high contribution of Api m 1 (19.6%) and Api m 10 (14.4%), medium contribution of Api m 2 (7.6%), Api m 3 (7.2%) and Api m 5 (8.9%) and low contribution of Api m 4 (2%). The latter was commercial availability of the molecules. The paragraph that describes the performing of skin tests has been modified

  1. There is no information whether the molecules were recombinant or native. Please add this information as it is essential for the discussion of the results.

Ad 5. Thank you for your comment. Molecules were recombinant. It was corrected in the text

  1. The tables are far to extensive and their number should be reduced. All tables and figures need extensive revision (see minor comments for concrete recommendations).

Ad 6. Thank you for your comment. It was corrected

  1. It is important to correctly distinguish “allergy” from “sensitization” throughout the entire manuscript. For example in line 79, the authors should say “sensitization profile”

Ad 7. Thank you for your comment. It was corrected in the text

  1.  

The manuscript needs careful revision by an English native speaker. Many sentences are unnecessarily complex and there are many grammar and spelling mistakes.

  1. A section on limitations is missing.

Ad 8m 9.Thank you for your comment.  The manuscrpt has been double checked by a native English speaker. Grammar and spelling mistakes have been corrected .The limitation section was added.

Minor comments:

  1. Lines 9 ff Abstract: The abstract should start with a short introductory sentence on the general background.

Tad 1.hank you for your comment. The introductory sentence in the abstract has been updated as follows: Hymenoptera venom (HVA) allergy is the most common cause of anaphylaxis in adults

  1. Abbreviations should be fully described when first mentioned. After that, only the abbreviation should be mentioned. It is unnecessary to introduce abbreviations (like in the first sentences of the introduction) if these terms are then not used later in the manuscript. There should be a thorough list of abbreviations as not all of them have been introduced and are simply used, so the reader does not know what they stand for.

Ad 2. Thank you for your comment. It was corrected in the text.The list of abbreviations was added.

Thank you for your comment. The table was added.

  1. Recommendations for all tables: headers are far too long. Use abbreviations and remove non-essential information. Pay attention to correct spelling (e.g. in table 1 it should be severity of allergic reaction) and proper wording (e.g. sex is usually female/male and not w/m). Median should not be abbreviated Me as the word “Mean” also starts with the same letters. It needs to be clear for every number in a table what it described, including indication of units (e.f. what are the values in table 1 column 4?).

Ad 3. Thank you for your comment. Tables were corrected.

  1. Table 1 should be removed. The results can be mentioned in the text very shortly. Instead, the information on allergenic molecules should rather be sumarized in a table than in a long text passage.  

Ad 4. Thank you for your comment. Table 1 was removed. We added new table with information on allergenic molecules

  1. Table 2 should be moved to the e-repository. Table 3 should be removed, this information is not essential for the manuscript.

Ad 5. Thank you for your comment. Table 3 was removed.

  1. Table 4 should be moved to the e-repositoryFigure 1: as some sensitization profiles are repeated in several patients this figure should be condensed. Instead of indicating 46 profiles, only the 21 distinct profiles should be shown and a column should be added showing how many patients had the respective profile. Ideally the profiles should then be sorted top down according to their prevalence.

Ad 6. Thank you for your comment. Figure 1 was modified..

  1. Table 8 : the table should be deleted. Data here collected should be explained in the results.

Ad 7. Thank you for your comment. Table 8 was removed.

  1. References: make sure to use correct reference styles. For example, the journal in Ref 9 is not indicated correctly.

Ad 8. Thank you for your comment. It was corrected.

  1. Tryptase results are not discussed. As they fit previous knowledge, no large discussion is needed, but they should at least shortly be mentioned in the context of other researchers´work.

Ad 9. Thank you for your comment. It was included in text. High concentration of sBT is a known, unmodified risk factor for the severity of the post-stinging reaction and for VIT failure. In the presented study, the results of tryptase concentration are compatible with the previous reports. We observed higher levels of tryptase in patients with SYS IV systemic reactions (p = 0.007) and in the group of patients with complications during the VIT -UR induction phase (p = 0.019) and in the course of whole process of immunotherapy (p = 0.03).

  1. How do the authors explain the rather high rate of patients with no IgE response at all or only reacting to bee venom extract? Please take cross-reactivity into consideration and discuss this aspect.

Ad 10. Thank you for your comment. It was included in text. The high percentage of patients with confirmed allergy to the extract and unconfirmed allergy to the tested venom allergen components may have two explanations. First, the concentration of sIgE against the venom components may be too low in relation to the sensitivity of current methods of their detection. Second, the lack of sIgE against the test molecules may reflect an allergy to "new" as yet unidentified "smaller" bee venom allergens (eg Api m 11 or Api m 12). In these patients, the potential efficacy of VIT may be lower due to the lack of sufficient allergenic molecule in the vaccine. Determining sIgE against higher numer of molecules increases the diagnostic value of the procedure.

Reviewer 2 Report

 However, there are some points listed below need to be clarified.

Specific comments

  1. The grammer of article is not good. It should be read by a native speaker.
  2. The article was not well planned and written.
  3. In abstract: What are sBT and VIT-UR? Please first define them than use abbreviation.
  4. In the introduction authors stated that venom allergy is the most common cause of anaphylaxis in adults and they referred to a relatively old article. Please check the literature, since currently the most common cause of anaphylaxis is drug hypersensitivity in adults.
  5. In the method ‘…the concentration of sIgE against 5 HBV molecules (Api m 1 (n = 64) and in 46 individuals also the sIgE against Api m 2, Api m 3, Api m 5 and Api m 10.’ Please complete the sentence.
  6. The study population was compared to an another group named entire? (line 89) or reference ? (in table legend). Please define your control group well in the method section and explain why to use such a group in brief.
  7. In table 1 ‘ssIgE’ remove one ‘s’, explain the abbreviation at the bottom of the table and ‘Severity off alergic reaction’ rewrite the ‘off’.
  8. Did you use only intradermal test as a skin test? Why did not use first prik test? Please explain.
  9. ‘The testing technique was complaint with the recommendations of ENDA / EAACI’ what did you mean with ‘complaint’ here? Did you mean the technique was performed ….?
  10. In line 99 ‘bubble’ is not suitable Word please use enduration.
  11. Table 2 should not be in the main article it can be given as supplement.
  12. Line 115, sBT? First write it fully then use abbreviation.
  13. In line 118 did you mean history of …reaction? If so ‘past’ is not a suitable word.
  14. In the method section it eas understood that muller severity classification was used for both field sting reaction and VIT reaction, but in the legend of table 4 it was written just for field sting. Please correct the wrong one.
  15. The statistical analysis section is not enough. Even the information about statistical significance level was not mentioned.
  16. In any study, in the result section the population should be clearly defined by giving demographic and clinical features for readers at first. Here, results started with directly molecular findings. Also you used clinical features in the article. For example, in line 161 you gave a finding about cardiovascular involvement. That’s not appropriate. First you should give detail about reaction involvement than you can use them in comparisons etc.
  17. What is SYS? Write it in full version at first.
  18. Please use small ‘n’ not ‘N’.
  19. Legend table 5 ‘Concentrations of sIgE antibodies against bee venom allergen molecules and the concentration of tryptase in the group of patients with bee venom allergy and reaction SYS IVº and SYS Iº, IIº or IIIº’  is not suitable ‘Comparison of serum levels of spesific IgE against bee venom allergen molecules and basal tryptase among patients with reaction SYS IV and SYS I,II or III’  would be better.
  20. What did you mean with ‘7 allergic side effects’? in 7 patients? Or in 7 injections?
  21. What is ‘median IgE levels of IgE antibodies’?
  22. In table 7 did you give the result of maintenance phase of VIT? If it was table 6 and 7 should be unit.
  23. In line 210, What is ‘post-stinging reaction’? did you mean field-sting? If so, you did not give any information about field sting reactions in results. Or Did you mean the reaction in the history? Please define this well in method section.
  24. The authors did not include a conclusion in the article.

Author Response

Thank you very much for your comment..

1.The grammer of article is not good. It should be read by a native speaker.

Ad 1. Thank you very much for your comment. The manuscript has been checked by the a native English speaker. The grammar and spelling mistakes have been corrected.

  1. In abstract: What are sBT and VIT-UR? Please first define them than use abbreviation.

Ad 2. It was corrected. Thank you for your remarks. The abstract was corrected. We added the sentence about the objective of the study. The main hypothesis was to determine “whether specific profiles of molecular IgE responses are associated with an increased risk of systemic side effects or treatment inefficacy?” The hypothesis has been specified in both the abstract and the introduction to the paper. Fragments of the discussion have been revised so that the study’s results could be clearly commented on.

  1. In the introduction authors stated that venom allergy is the most common cause of anaphylaxis in adults and they referred to a relatively old article. Please check the literature, since currently the most common cause of anaphylaxis is drug hypersensitivity in adults.

Ad 3. Yes. We agree. The elicitor profile of anaphylaxis is age-dependent and varies between different geographic areas. According to WAO document the main elicitors of anaphylaxis are: drugs, insect venom and food (Cardona et al. World Allergy Organization Journal (2020) 13:100472). This paper is included in the References section.

  1. In the method ‘…the concentration of sIgE against 5 HBV molecules (Api m 1 (n = 64) and in 46 individuals also the sIgE against Api m 2, Api m 3, Api m 5 and Api m 10.’ Please complete the sentence.

Ad 4. Thank you for your comment. SIgE antibodies against Api m 1 were assessed in 64 individuals. SIgE against a panel of molecules Api m 2, 3, 5,10 were assessed in 46 of them. For technical reasons, sIgE assessment against the entire panel in the whole study group was not possible. Even so, we decided to present the results of the assessment of the presence of sIgE against  Api m 1 in 64 individuals in order that the analysis of the involvement of sIgE against this molecule could be of carried out on a larger sample of subjects. Following the reviewer’s suggestion, the group of 18 patients (sIgE against Api m 1 alone) was compared with the group of 46 patients (sIgE against the whole molecular panel) as done in table 1 for the study population vs. the general patient population. No significant differences between the two groups were found.  Now the senstence reads: “…the concentration of sIgE against 5 HBV molecules (rApi m 1 (n = 64) and in 46 individuals also the sIgE against rApi m 2, rApi m 3, rApi m 5 and rApi m 10 were determined.”

  1. The study population was compared to another group named entire? (line 89) or reference ? (in table legend). Please define your control group well in the method section and explain why to use such a group in brief.

Ad 5, Between 2010 and 2019 VIT with bee venom was carried out in 232 patients. A group of 64 patients was recruited for the present study for whom sIgE assessments against rApi m 1 were performed. For the subgroup of 46 patients (46 out of the initial 64) assessments of sIgE against rApi m 2, 3, 5, i 10 were additionally performed. Clinical and immunological features were compared between the entire cohort (232 patients) and the study subgroups (n=64) and (n=46); no significant differences were observed. In conclusion, the subgroup (n=46) proved to be credible for assessing the parameters investigated in the study.

  1. Did you use only intradermal test as a skin test? Why did not use first prik test? Please explain.

Ad 6. In our VIT center, likewise in others we know, we do not use the prick test for diagnosis due to its low sensitivity, and instead we use IDT test which is both safe and more sensitive. The sensitivity of prick test with venom vs IDT test in diagnosis of allergies to bee and wasp venoms is 85% vs 96 % (wasp venom) and 73% vs 91% (bee venom) (Sturm et al).

  1. ‘The testing technique was complaint with the recommendations of ENDA / EAACI’ what did you mean with ‘complaint’ here? Did you mean the technique was performed ….?

Ad 7. The sentence has been corrected. Now it reads: ”The technique  and interpretation of the test was compatible with the guidelines and recommendations of ENDA and EAAACI.”

  1. In line 99 ‘bubble’ is not suitable Word please use enduration.

Ad 8. Thank you for the remark. A correction has been introduced. 

  1. Table 2 should not be in the main article it can be given as supplement.

Ad 9. Table 2 has been moved to the supplement.

  1. Line 115, sBT? First write it fully then use abbreviation.

Ad 10, sBT has been disabbreviated in the text.

  1. In line 118 did you mean history of …reaction? If so ‘past’ is not a suitable word.

Ad 11. Thank you. “past history” has been replaced with “history of reaction.”

  1. In the method section it eas understood that muller severity classification was used for both field sting reaction and VIT reaction, but in the legend of table 4 it was written just for field sting. Please correct the wrong one.

Ad 12. Mueller’s scale was used in the study both for the classification of the severity of allergic reactions and for the assessment of the severity of adverse events during VIT. Correction has been made in the text.

  1. The statistical analysis section is not enough. Even the information about statistical significance level was not mentioned.

Ad 13. The section „Statistical analysis” has been improved.

  1. In any study, in the result section the population should be clearly defined by giving demographic and clinical features for readers at first. Here, results started with directly molecular findings. Also you used clinical features in the article. For example, in line 161 you gave a finding about cardiovascular involvement. That’s not appropriate. First you should give detail about reaction involvement than you can use them in comparisons etc.

Ad 14. Thank you. The study group has been described in more detail in the section ”Results.”

  1. What is SYS? Write it in full version at first.

Ad 15, The abbreviation SYS (systemic reactions) was disabbreviated when it was used in the text for the first time.

  1. Please use small ‘n’ not ‘N’.

Ad 16. Thank you. Correction has been made.

  1. Legend table 5 ‘Concentrations of sIgE antibodies against bee venom allergen molecules and the concentration of tryptase in the group of patients with bee venom allergy and reaction SYS IVº and SYS Iº, IIº or IIIº’  is not suitable ‘Comparison of serum levels of spesific IgE against bee venom allergen molecules and basal tryptase among patients with reaction SYS IV and SYS I,II or III’  would be better.

Ad 17. Correction has been made.

  1. What did you mean with ‘7 allergic side effects’? in 7 patients? Or in 7 injections?

Ad 18, Adverse events occurred in 7 patients. If such events recurred a number of times in the same patient, the most severe of them was taken into account.

  1. What is ‘median IgE levels of IgE antibodies’?

Ad 19. It is the median of IgE in serum.

  1. In table 7 did you give the result of maintenance phase of VIT? If it was table 6 and 7 should be unit.

Ad 20. Correction has been made.

  1. In line 210, What is ‘post-stinging reaction’? did you mean field-sting? If so, you did not give any information about field sting reactions in results. Or Did you mean the reaction in the history? Please define this well in method section.

Ad 21. The phrase is defined in the ”Method” section.

  1. The authors did not include a conclusion in the article.

Ad 22. The „Discussion” section has been supplied with conclusions

This manuscript is a resubmission of an earlier submission. The following is a list of the peer review reports and author responses from that submission.